# Chiral orbital lasing in a twisted bilayer metasurface

Mingjin Wang [1,2,3,4,9], Nianyuan Lv[5,9], Zixuan Zhang [5,9], Ye Chen [5], Jiahao Si[1], Jingxuan Chen [1], Chenyan Tang[1], Xuefan Yin[5], Zhen Liu[5], Dongxu Xin[1], Zhaozheng Yi[1], Wanhua Zheng [1,2,3,4] ✉, Yuri Kivshar [6,7] ✉ & Chao Peng [5,8] ✉

Chirality is a fundamental concept in physics, core to many phenomena in nonlinear optics, quantum physics and topological photonics. Photons are intrinsically chiral when carrying spin angular momentum, whereas orbital angular momentum can induce chirality when photons interact with structures with broken mirror symmetry. Here, we observe orbital chiral lasing from a twisted bilayer photonic structure, by leveraging its inherent structural chirality. We design and fabricate a Moiré-type optical structure consisting of two semiconductor membrane metasurfaces. By optically pumping the twisted bilayer, we achieve single-mode lasing over a broad spectral range of 250 nm. The lasing emission exhibits chiral orbital characteristics, arising from helical and non-Hermitian couplings between collective guided resonances rotating clockwise and counterclockwise and confirmed by polarization-resolved imaging and self-interference measurements. The observed chiral orbital lasing from a twisted photonic structure can contribute to diverse applications of chiral light in diagnostics, optical manipulation and communication with light.

The emerged field of twistronics is underpinned by the pioneering observations that twisted stacking of two-dimensional materials can change their electric properties from being non-conductive[1,2] to superconductive[3–10], thus opening up avenues for advanced material design and inspiring research into their photonic counterparts. In twisted bilayer photonic systems[11–15], interlayer coupling acts as an additional degree-of-freedom to stipulate unusual phenomena such as strong light localization[16–18], enhanced optical nonlinearity[19,20], and induced circular dichroism[12,21], paving the way towards applications[22,23]. Among rich physics ranging from magic angles[24] to flatbands[25–29], a particularly intriguing characteristic of twisted bilayer systems lies in their intrinsic chirality[12,30,31], which refers to an object or a system that cannot be superimposed with its mirror image. It was

reported that a single-layer slanted metasurface can give rise to intrinsic chirality, represented by circularly polarized states with unique handedness known as spins for chiral lasing[32]. Despite this spin chirality, exciting advances have also been made in orbital chirality for light emission and lasing, typically by employing micro-ring cavities[33–35], helical phase plates[36,37], photonic crystals[38,39] and metasurfaces[32,40,41]. Orbital chirality of photons can be understood as a twisting motion of an electromagnetic field as it travels through space, presented by the sign of optical modes' orbital angular momentum (OAM). However, due to the law of reciprocity, optical modes in a vertical mirrored structure are always twofold-degenerate unless additional perturbations, such as helical geometry, gain, or loss, are introduced to break chiral symmetry[42–44]. In contrast, light traveling in

[1]Laboratory of Solid State Optoelectronics Information Technology, Institute of Semiconductors, CAS, Beijing, China. [2]Center of Materials Science and Optoelectronics Engineering, University of Chinese Academy of Sciences, Beijing, China. [3]Hangzhou Institute for Advanced Study, University of Chinese Academy of Sciences, Hangzhou, China. [4]College of Future Technology, University of Chinese Academy of Sciences, Beijing, China. [5]State Key Laboratory of Photonics and Communications, School of Electronics & Frontiers Science Center for Nano-optoelectronics, Peking University, Beijing, China. [6]Nonlinear Physics Centre, Research School of Physics, Australian National University, Canberra, ACT, Australia. [7]Department of Physics, The University of Hong Kong, Hong Kong, China. [8]Peng Cheng Laboratory, Shenzhen, China. [9]These authors contributed equally: Mingjin Wang, Nianyuan Lv, Zixuan Zhang. ✉e-mail: whzheng@semi.ac.cn; yuri.kivshar@anu.edu.au; pengchao@pku.edu.cn

twisted bilayer systems naturally couples in a non-Hermitian and helical manner, which could lead to collective oscillations possessing an intrinsically determined orbital chirality, distinct from that in single-layer systems. Consequently, twisted bilayer systems could be harnessed for on-chip chiral lasers—a long-sought goal in photonic research.

Here, we report on the observation of intrinsic orbital chiral lasing at telecom wavelengths using twisted bilayer metasurfaces. Specifically, we design a twisted photonic structure comprising two rotated metasurface membranes to form a Moiré-type lattice and employ a gain-guided mechanism to confine light. In such a system, isotropic geometry and dispersion hybridize bulk guided resonances, leading to a set of twofold degenerate collective guided resonances (CGRs) in each membrane that rotate in clockwise (CW) and counterclockwise (CCW) directions, respectively. Aided by the inherent chirality and non-Hermitian physics in the twisted photonic structure, the CW and CCW rotating modes interact to create non-Hermitian degeneracy, thus allowing one orbital chiral mode to prevail in the lasing. We further develop a wafer bonding process to fabricate the sample and achieve experimentally stable single-mode lasing over a broad spectral range of 250 nm at the lasing threshold of 73 kW/cm². The lasing emission exhibits an intrinsic orbital chiral nature as predicted, showing as a doughnut pattern in real space that carries a phase vortex with OAM quantum number $l = 1$, confirmed by polarization-resolved imaging and self-interference features.

## Results

### Principle and design

We start with a schematic of a twisted photonic structure as shown in Fig. 1a, which comprises two stacked and twisted metasurfaces with a thickness of $h \sim 620$ nm, separated by a gap of $g \sim 100$ nm, patterned on an epitaxial wafer with InGaAsP multiple quantum wells (MQWs) to provide optical gain in the telecom C-band around 1550 nm (details are provided in Supplementary Section 1). The upper and lower metasurfaces feature the same circular air holes arranged in square lattices, with a period of $a \sim 540$ nm and a radius of $r \sim 150$ nm. We numerically calculate the band structure of the single metasurface unit-cell near the 2nd order $\Gamma$ point (Fig. 1b), identifying four transverse electric

bands from TE-A to TE-D that lie in the continuum and radiate towards the out-of-plane direction, indicating that such a system is inherently non-Hermitian[45]. Nevertheless, the modes on the TE-A band still exhibit high quality factors ($Q$s) due to the presence of a symmetry-protected bound state in the continuum (BIC) at the $\Gamma$ point[46]. These modes are vertically confined in the metasurface membrane and transversely extend throughout an infinite area with a specific Bloch wavevector $\mathbf{k}$, and we refer to them as bulk guided resonances[47]. As illustrated in Fig. 1c, the TE-A band's dispersion is quite isotropic in momentum space, showing circular iso-frequency contours near the $\Gamma$ point.

Next, we rotate the upper metasurface membrane at an angle $\theta$ and place it over the lower one to form a twisted photonic system. This twist generates a Moiré pattern, leading to a supercell with a real-space size $L = Na$ (Fig. 1d). Consequently, the supercell occupies only a fraction of the reduced Brillouin zone (BZ) of the unit cell, which spans $[-\pi/L, \pi/L]$ in reciprocal space (green box, Fig. 1e). Moiré patterns emerge at specific twist angles $\theta$, referred to as magic angles, which determine the size of the supercell as $N = 1/(\sqrt{2}\sin(\theta/2))$[48,49]. This relationship implies that a larger twist angle results in a smaller supercell, and vice versa. Notably, diffraction in Moiré lattices causes out-of-plane leakage to degrade the $Q$s. In our design, we select $\theta = 22.62°$, corresponding to a supercell of $5/\sqrt{2}a \times 5/\sqrt{2}a$, ensuring that only a few diffracted waves fall within the light cone. Additionally, out-of-plane leakage can be further minimized by incorporating off-$\Gamma$ BICs by tuning membrane thickness. The TE-A band maintains quadratic and isotropic dispersion within this small-size supercell (Supplementary Fig. S1), which distinctly contrasts with the flat-band effect reported in the literature[24]. It is also important to highlight that such a twisted photonic structure is inherently chiral, meaning it cannot be superimposed onto its mirror image[12]. More details about the supercell design, radiation suppression, and parameter optimization are provided in Supplementary Sections 2, 3.

Further, we introduce an optical cavity to confine the bulk guided resonances to a finite region for collective oscillation[50,51]. To achieve this, we employ a gain-guided mechanism to create an effective cavity. Specifically, when the MQW material is optically pumped (Fig. 2a), carrier diffusion leads to spatial inhomogeneities in the gain, which in turn modify the refractive index for light confinement[52]. This variation

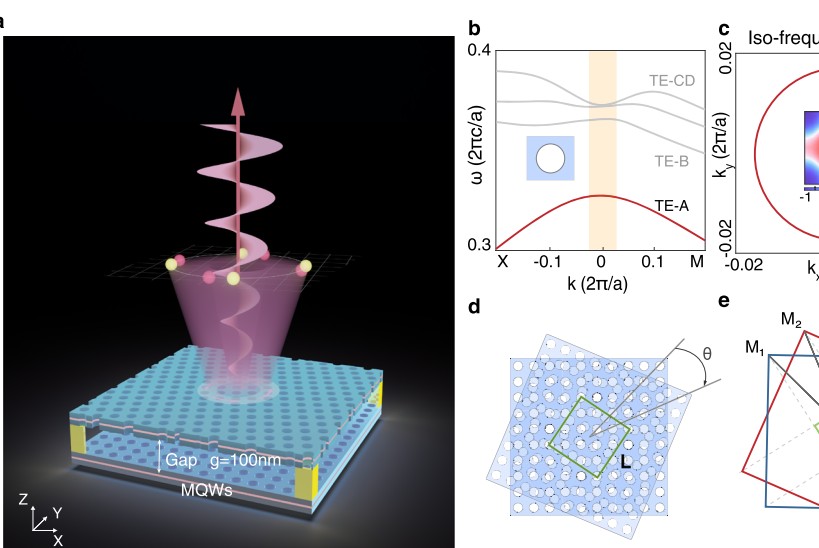

**Fig. 1 | Structure for intrinsic orbital chiral lasing. a** Schematic of twisted photonic structure. The system comprises a pair of twisted and bonded metasurface membranes with InGaAsP MQWs separated by a gap distance of 100 nm. **b** Bandgap structure of the unit cell (inset) for a single metasurface with $C_4$ lattice near the 2nd-$\Gamma$ point, where the TE-A band exhibits a high quality factor. **c** Two-dimensional visualization of the TE-A band's dispersion, showing a circular iso-frequency contour near the $\Gamma$ point. The inset illustrates the bulk TE-A mode's field distribution ($H_z$) at the $\Gamma$ point. **d** Real space schematic of the Moiré-type lattice with a twisted angle of $\theta$ creates a supercell in the size of $L$. **e** Reciprocal space representation of the Moiré-type lattice, the red and blue boxes show the reduced Brillouin zone of the unit cell for the single metasurface, and the green box denotes the reduced Brillouin zone of the Moiré supercell.

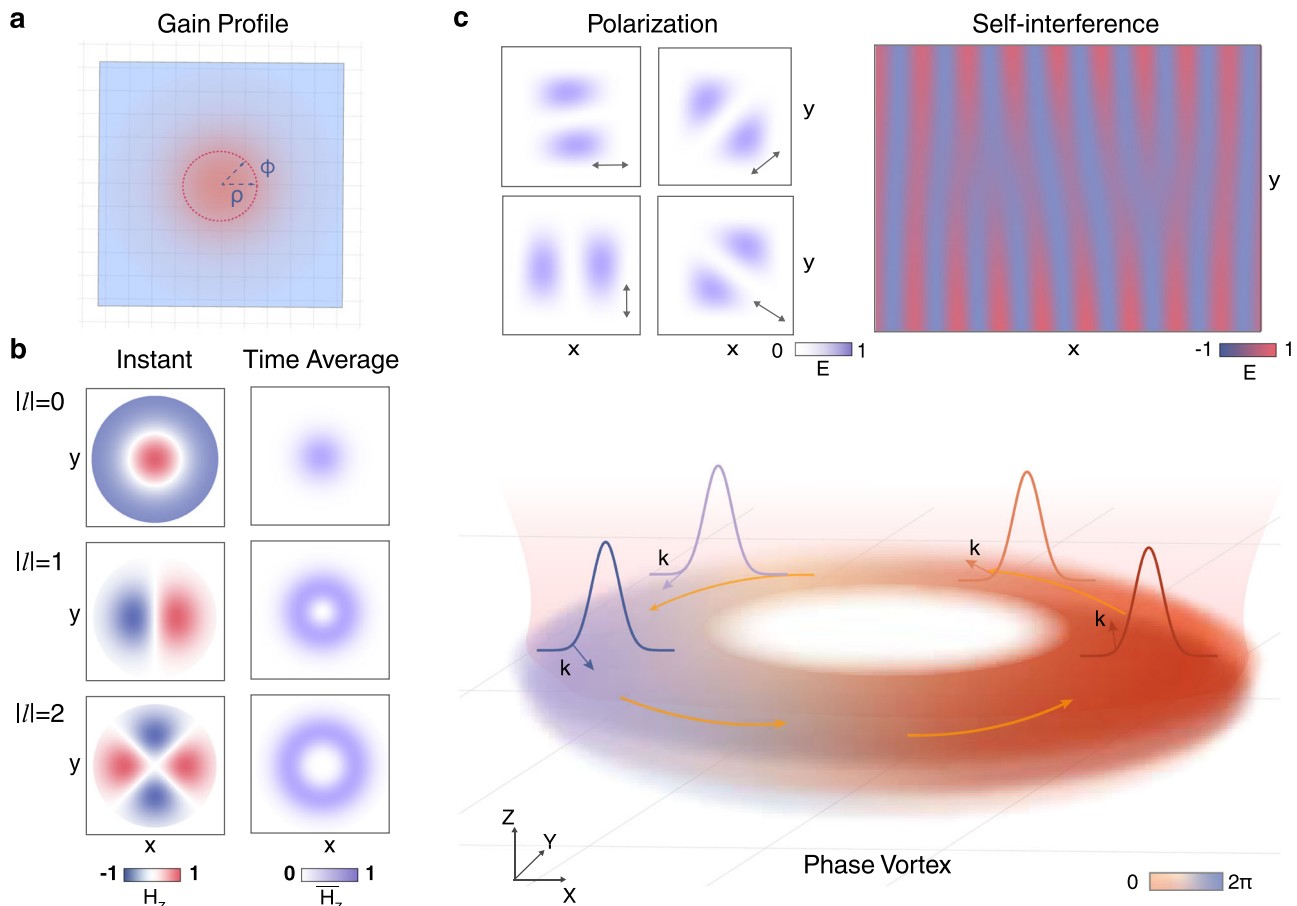

**Fig. 2 | Collective oscillations and emission in gain-guided metacavity.**
**a** Schematic of the gain-guided cavity with its refractive index spatially following the carrier density distribution. The red circle denotes the rotational-invariant pumped region. **b** Instantaneous and time-averaged field distributions of the collective guided resonance modes of $|l| = 0, 1, 2$ with $p = 0$. All modes except for $|l| = 0$ are twofold degenerate due to reciprocity. These modes carry nontrivial orbital angular momentum (OAM) and exhibit doughnut patterns in the time-averaged field. **c** Schematic of one collective mode $l = 1$, $p = 0$ rotating along CCW direction in real space (lower panel), which diffracts to out-of-plane radiation, creating two lobes with equal intensity along any polarized direction (upper-left panel) and a fork-like self-interference pattern (upper-right panel).

can be approximated by $n(\rho) = n_1(0) - n_2\rho^2$, where $\rho$ is the radial position and $n_1$, $n_2$ are coefficients that depend on the material and pump power. The gain-induced refractive index evolves continuously according to the carrier density distribution that follows the profile of the pump beam. As a result, a circular pump beam naturally forms a rotationally symmetric cavity regardless of the twist angle. That is, the gain-guided effective cavity remains achiral.

As demonstrated[53], trapping guided resonances with isotropic dispersion into a geometrically isotropic region can give rise to a set of CGRs that preserve continuous rotational symmetry, appearing as paired CW and CCW modes rotating in real space even without explicitly defined physical paths. We denote these CGRs' wavefunctions as $\left|\psi_{CW, CCW}^{u, l}\right\rangle$, where the superscript indicates the metasurface position and the subscript denotes the rotating direction. Given that each metasurface is identical and achiral, these modes share the same complex eigenfrequency $\Omega_0$ governed by $\mathbf{H_0}\left|\psi_{CW, CCW}^{u, l}\right\rangle = \Omega_0\left|\psi_{CW, CCW}^{u, l}\right\rangle$, where $\mathbf{H_0}$ represents the Hamiltonian of a single metasurface membrane[54]. In our gain-guided cavity featuring a quadratic index changing, $\left|\psi_{CW, CCW}^{u, l}\right\rangle$'s envelopes take the form of Laguerre-Gaussian functions $\Psi_m(r_z)$ (details are provided in Supplementary Section 4), characterized by quantum numbers $l$ and $p$ that correspond to the quantization in azimuthal and radial directions, respectively. The left panel of Fig. 2b presents the instantaneous field distributions of the

collective modes for $|l| = 0, 1, 2$ with $p = 0$, where each mode exhibits twofold degeneracy due to the absence of chirality in both geometry and dispersion. The right panel of Fig. 2b illustrates the time-averaged field distributions, showing that, apart from the lowest mode $|l| = 0$, the average field profiles form doughnut-shaped patterns in real space. Additionally, the metasurfaces diffract CW or CCW modes into vertical radiation, producing a phase vortex beam (Fig. 2c). When observed through a linear polarizer, the diffraction results in two lobes of equal intensity appearing along the direction perpendicular to any polarizer's major axis. By splitting and recombining the radiated beam for self-interference, a fork fringe is expected, serving as a distinct signature of chiral emission (upper panel, Fig. 2c).

We further consider a compound bilayer system, importantly, showing that the structure's intrinsic chirality naturally breaks the twofold degeneracy in each membrane via intra and inter-layer couplings. For example, if we examine $\left|\psi_{CW, CCW}^{u}\right\rangle$ residing in the upper metasurface, the lower one can be regarded as an asymmetric scatterer in real space that turns the upper one to be chiral, described by a perturbed Hamiltonian $\Delta\mathbf{H}$ that introduces helical diffraction strengths. Consequently, the lower metasurface facilitates both CW → CCW and CCW → CW coupling paths in the upper one but with different weights, which we denoted as intra-layer couplings. Besides, the overlap of evanescent waves along the vertical direction also gives rise to inter-layer couplings. Taking into account radiation leakage as well as material gain or loss, the coupling coefficients are generally

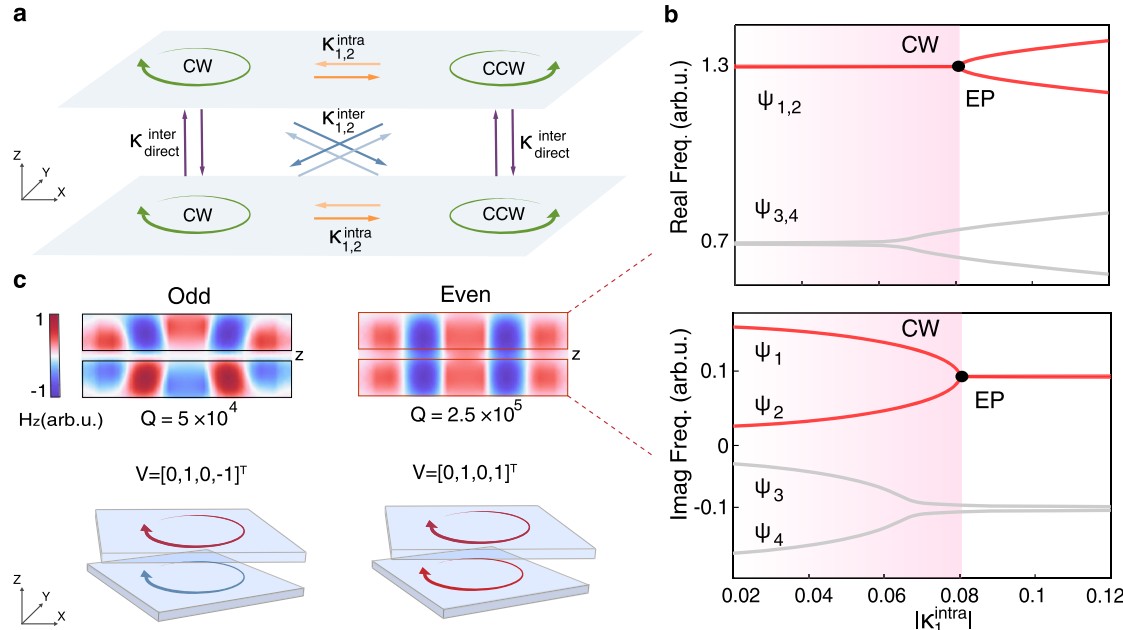

**Fig. 3 | Coupling in the twisted bilayer system. a** Coupling scenario of four unperturbed single-layer modes $\left|\psi_{CW,\,CCW}^{u,\,l}\right\rangle$, illustrating three distinct coupling paths as intra-layer cross-coupling ($\kappa_{1,2}^{\text{intra}}$, orange arrows), inter-layer cross-coupling ($\kappa_{1,2}^{\text{inter}}$, blue arrows), and inter-layer direct-coupling ($\kappa_{\text{direct}}^{\text{inter}}$, purple arrow). The coupling coefficients are complex due to the non-Hermiticity. **b** Complex bands of hybridized modes $\left|\psi_{1-4}\right\rangle$ are tuned by varying the intra-layer cross-coupling strength $\kappa_1^{\text{intra}}$. An exceptional point (EP) emerges at $\kappa_1^{\text{inter}} = -\kappa_1^{\text{intra}}$, where the eigenstates collapse. At the EP, the system reaches the highest degree of chirality, showing CW rotation existing in both layers. **c** Two representative chiral modes are generated through helical and non-Hermitian couplings with eigenvectors $\mathbf{V} = [0, 1, 0, -1]^T$ and $\mathbf{V} = [0, 1, 0, 1]^T$, respectively. Here, the upper layer is fixed to CW rotating and their z-phases are 0 (red arrow) or $\pi$ (blue arrow). The mode $\mathbf{V} = [0, 1, 0, 1]^T$ corresponds to the configuration shown in (**b**).

complex values that represent non-Hermitian interactions. More details about helical and non-Hermitian couplings are presented in Supplementary Section 5.

To gain insight into the coupling dynamics in the twisted structure, we write an effective Hamiltonian $\mathbf{H}_{\text{eff}}$ by including these four interacting, radiative, single-layer CGR modes $\left|\psi_{CW,\,CCW}^{u,\,l}\right\rangle$ as schematically illustrated in Fig. 3a, in which the coupling paths are categorized into three distinct types, which are intra-layer cross-coupling ($\kappa_{1,2}^{\text{intra}}$, orange arrows), inter-layer cross-coupling ($\kappa_{1,2}^{\text{inter}}$, blue arrows), and inter-layer direct-coupling ($\kappa_{\text{direct}}^{\text{inter}}$, purple arrow). Here, the terminology "cross" and "direct" refer to the couplings occurring between modes with opposite or same rotating directions, respectively. Besides, the subscripts 1, 2 denote the couplings direction as 1:CCW → CW and 2: CW → CCW, respectively. As a result, the effective Hamiltonian $\mathbf{H}_{\text{eff}}$ can be written as a 4 × 4 coupling matrix, showing as an eigenvalue problem in the basis of single-layer wavefunctions $\mathbf{V} = [\left|\psi_{CCW}^u\right\rangle, \left|\psi_{CW}^u\right\rangle, \left|\psi_{CCW}^l\right\rangle, \left|\psi_{CW}^l\right\rangle]^T$:

$$\mathbf{H}_{\text{eff}}\mathbf{V} = \begin{pmatrix} \Omega_0 & \kappa_1^{\text{intra}} & \kappa_1^{\text{inter}}_{\text{direct}} & \kappa_1^{\text{inter}} \\ \kappa_2^{\text{intra}} & \Omega_0 & \kappa_2^{\text{inter}} & \kappa_{\text{direct}}^{\text{inter}} \\ \kappa_{\text{direct}}^{\text{inter}} & \kappa_1^{\text{inter}} & \Omega_0 & \kappa_1^{\text{intra}} \\ \kappa_2^{\text{inter}} & \kappa_{\text{direct}}^{\text{inter}} & \kappa_2^{\text{intra}} & \Omega_0 \end{pmatrix} \mathbf{V} = \Omega'\mathbf{V} \quad (1)$$

Here $\mathbf{H}_{\text{eff}}$ involves six independent complex parameters: $\Omega_0$, $\kappa_{\text{direct}}^{\text{inter}}$, $\kappa_{1,2}^{\text{inter}}$, $\kappa_{1,2}^{\text{intra}}$ from symmetry considerations. By solving this non-Hermitian eigenvalue problem, we obtain four hybridized modes $\left|\psi_{1-4}\right\rangle$ in the basis of $\left|\psi_{CW,\,CCW}^{u,\,l}\right\rangle$ (see Supplementary Section 6 for the details). Further, we identify four individual conditions for realizing exceptional points (EPs) as $\kappa_1^{\text{inter}} = \pm\kappa_1^{\text{intra}}$ or $\kappa_2^{\text{inter}} = \pm\kappa_2^{\text{intra}}$, where the eigenstates collapse at non-Hermitian degeneracy. As an example, we

discuss the EP at $\kappa_1^{\text{inter}} = -\kappa_1^{\text{intra}}$ and calculate the complex eigenfrequencies as $\kappa_1^{\text{intra}}$ varies (Fig. 3b), revealing that the eigenstates $\left|\psi_{1-4}\right\rangle$ fall into two distinct branches, with one possessing higher loss. The EP is found on the upper branch, at which the eigenvector taking the form $\mathbf{V} = [0, 1, 0, 1]^T$ with the highest degree of chirality, indicating that the collective mode merely rotates in the CW direction in both upper and lower layers with the aligned phases. When $\kappa_1^{\text{intra}}$ deviates from the EP within the shaded region, the real parts of the eigenfrequencies remain nearly degenerate. However, the differences in their imaginary parts (namely gains) can leverage one mode to dominate in lasing competition, ultimately prevailing over the others for single-mode lasing action. Noteworthy that, near the EPs, the degree of chirality slightly degrades but remains relatively high to enable the observation of chiral emission (Supplementary Fig. S5).

The EP condition $\kappa_1^{\text{inter}} = -\kappa_1^{\text{intra}}$ signifies a balanced interplay between intra and inter-layer coupling, manifesting that a combined symmetry of non-Hermiticity breaks the reciprocity. Notably, remaining EPs can also support chiral emission, with their corresponding eigenvectors given by $\mathbf{V} = [0, 1, 0, -1]^T$, $[1, 0, 1, 0]^T$, and $[1, 0, -1, 0]^T$. These eigenvectors differ in their transverse rotational direction and vertical phase relationships. Specifically, the modes can be either in or out-of-phase in the z direction, while their rotating direction can be either CW or CCW. The balance between $\kappa_{1,2}^{\text{inter}}$ and $\kappa_{1,2}^{\text{intra}}$ determines these characteristics as illustrated in Fig. 3c. Interestingly, the EPs in a system of two stacked dielectric metasurfaces have been theoretically investigated from the perspective of asymmetric losses[55], further highlighting the rich physics of non-Hermitian bilayer systems.

### Sample fabrication and experimental setup
To verify the principle of twisted photonic structure for orbital chiral lasing, we fabricate the samples by bonding two metasurface membranes from an InGaAsP MQW wafer on an InP substrate. The two sheets of square latticed metasurface are patterned at a twist angle of

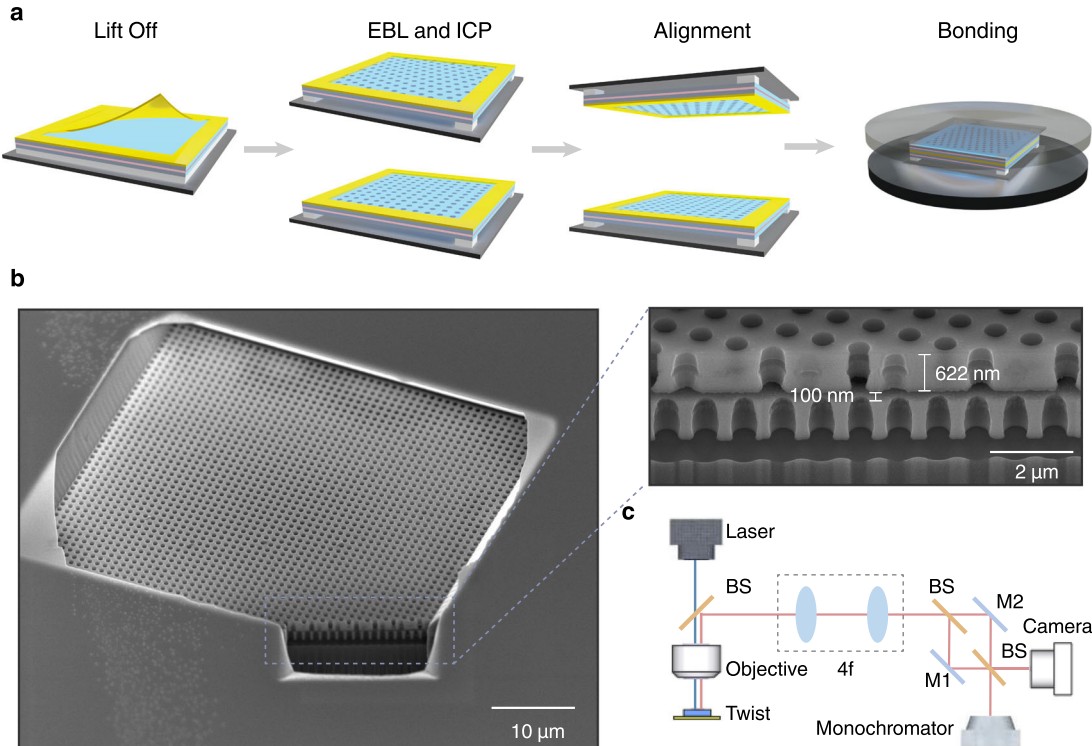

**Fig. 4 | Sample fabrication and measurement setup. a** Fabrication processes for twisted bilayer metasurfaces, that include metal deposition and lift-off, e-beam lithography, inductively coupled plasma etching, alignment by marks, and lower-temperature bonding. **b** Scanning electron microscope (SEM) images of the fabricated sample that was cleaved by focused ion beam (FIB). The zoom-in image shows the sample consisting of the upper and lower metasurfaces overlaid at a twisted angle, with their structural parameters being measured and marked. The air holes are slightly damaged during the wet-etching process to remove the upper cladding for the observation. **c** Schematic of the confocal 4*f* measurement setup. BS beam splitter, M gold-coated mirror.

22.62° within the same square-shaped metal out-frame for supporting and alignment. As shown in Fig. 4a, we first deposit and lift off a thin layer of titanium/gold with a thickness of 50 nm, which can create an air gap of 100 nm after bonding to adjust the coupling strength between two metasurfaces. Then, we fabricate each metasurface that consists of 50 × 50 air-holes at a lattice constant of $a = 540$ nm by using e-beam lithography (EBL) and induced plasma etch (ICP). The air-hole depth and radius are given by $h = 622$ nm and $r = 153$ nm, respectively. We further undercut the scarified layer underneath the MQW layer by using wet etching, to ensure $z$-mirror symmetry in a single sheet. Next, the two metasurfaces are aligned via prepared marks that are visible under the microscope of the wafer bonding machine, and then we bond them through a low-temperature annealing process (see Methods for details). We present the overall and detailed side views of the fabricated sample that is cleaved by a focused ion beam (FIB) (Fig. 4b), clearly showing that the upper and lower metasurfaces sandwich an air gap and align with an evident twist angle.

We use a confocal measurement system to characterize the lasing of the twisted photonic structure (Fig. 4c). A pulsed pump laser operated at a wavelength of 1064 nm was focused on the sample surface using a 20× objective lens to drive the lasing oscillation and chiral emission. The laser emission and the reflected pump beam are collected by the same objective lens, but the latter is filtered out by using a long-pass filter with a cutoff wavelength of 1300 nm. The laser emission is then imaged in real space by using a cascade 4*f* system. The laser emission pattern is captured by a complementary metal-oxide-semiconductor (CMOS) camera, while the lasing spectrum is recorded using a spectrometer. To identify the phase vortex associated with chiral emission, a Mach–Zehnder interferometer was constructed for self-interference, consisting of two beam splitters (BS) and two gold-coated mirrors (M1 and M2). The optical path difference was controlled by adjusting the angular orientation of M2. The interference patterns of lasing emission are ultimately captured by the CMOS camera. More details about the measurement setup can be found in Supplementary Section 7.

## Observation of intrinsic orbital chiral lasing

To experimentally observe the orbital chiral lasing behavior of the twisted bilayer structure, we optically pump the fabricated sample using the 1064 nm pulsed laser at room temperature. The pump beam has a Gaussian intensity profile in the transverse direction, showing as a circular spot of ~5.5 μm (10$a$) in diameter that preserves a rotational symmetry at arbitrary angles. Notably, the pump beam size is smaller than that of the metasurfaces' footprint (50$a$ × 50$a$). This size difference ensures that light confinement is primarily governed by gain-guided refractive index modification rather than the metasurfaces' outer geometric edges. More details are presented in Supplementary Section 8.

The establishing of lasing oscillation is observed as gradually increasing the pump power (Fig. 5a). We first record the emission spectrum with a 150 g/mm grating at a pump power of 70 kW/cm². The spectrum shows no evident resonant peaks upon the spontaneous emission background (upper panel, Fig. 5a). When the pump power reaches 73 kW/cm², a lasing threshold is achieved, evidenced by the emergence of a distinct peak amidst the spectrum (middle panel, Fig. 5a). Further increasing the pump power to 74 kW/cm² can suppresses spontaneous emission, leading to a well-defined single peak that features a single-mode lasing behavior spanning a broad spectral range of 250 nm from 1430 nm to 1680 nm (lower panel, Fig. 5a). Additionally, we record and plot the lasing power curve (Fig. 5b), highlighting three key data points (labeled A–C) corresponding to the

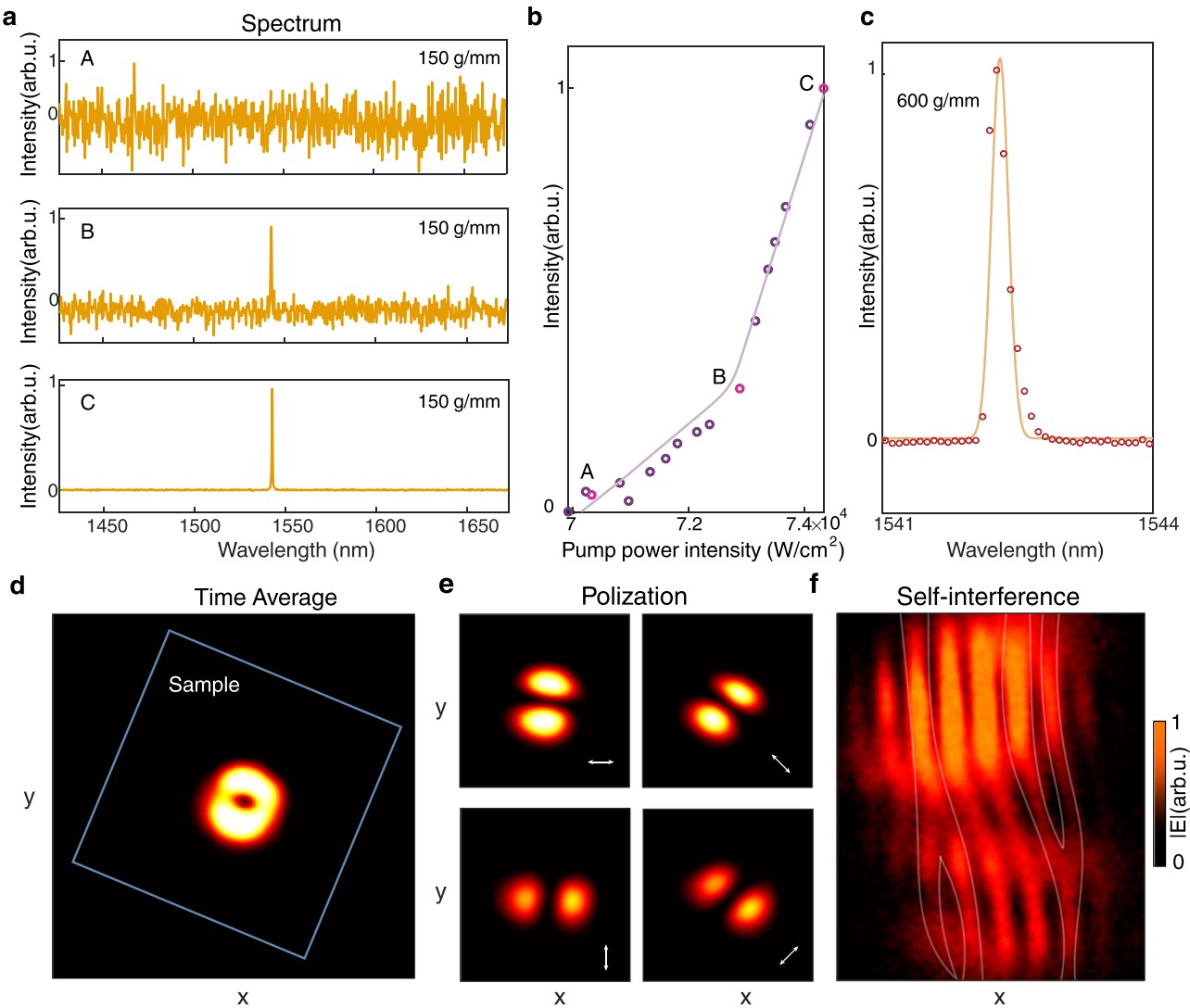

**Fig. 5 | Characterization of intrinsic orbital chiral lasing. a** Spectra of the lasing process when gradually increasing the pump power from 70 kW/cm² to 74 kW/cm². The lower panel (case C) shows the single-mode feature spans a broad spectral range of 250 nm. **b** Emission vs. pump power curve of the lasing process, which indicates the lasing is achieved at a threshold of 73 kW/cm². **c** Lasing spectrum measured using a high-resolution grating with 600 g/mm, revealing a distinct single peak and confirming single-mode operation. **d** Lasing beam pattern recorded by the CMOS camera, showing a doughnut shape in real space, the blue box indicates the physical boundary of the sample. **e** Polarization-resolved distribution of the lasing beam along 0°, 45°, 90°, and 135°, respectively. **f** The off-center self-interference pattern of the lasing beam, exhibiting two reversely oriented forks.

spectra in Fig. 5a. Notably, point B marks the power transition and represents the lasing threshold.

Further, we opt for a high-resolution grating (600 g/mm) to capture the lasing mode's spectral details. As we stated above (Fig. 3), four hybridized modes $|\psi_{1-4}\rangle$ evolve in lasing process, which might appear as multiple peaks in the emission spectrum if their real eigenfrequencies split. As presented in Fig. 3c, $|\psi_{1-4}\rangle$ comes as two distinct branches mostly aligning with the inter-layer coupling strength $\kappa^{inter}_{direct}$. Due to the twisted bilayer structure's inherent non-Hermiticity and strong inter-layer coupling from a small gap distance of $g \sim 100$ nm, the lower branch $|\psi_{3-4}\rangle$ separates ~5 nm from the high branch $|\psi_{1-2}\rangle$ and possesses considerable losses, thus they are not visible in our observation. Besides, the real frequencies of high branch modes $|\psi_{1-2}\rangle$ remain quite close, while their imaginary parts are different near the EP condition. As a result, we observe only one lasing peak rather than multiple closely spaced peaks upon the high-resolution spectrum, confirming that the twisted bilayer metasurface can support single-mode chiral lasing, despite the pump beam itself being achiral. More details are presented in Supplementary Section 9.

Next, we characterize the lasing emission from real-space imaging. a doughnut pattern is observed in real space (Fig. 5d), agreeing with our theory as laguerre-gaussian profiles in time-averaged measurements of any quantum number $|l| \neq 0$. the doughnut beam size is estimated to be ~5a, smaller than that of the pump beam size (~10a) and metasurfaces' footprint (~50a). when we shift the pump beam to a different position on the sample, the lasing beam follows the pump beam's new location while maintaining its doughnut shape (Supplementary Video), confirming the effectiveness of the gain-guided mechanism. to verify that the lasing doughnut beam is rotationally invariant, we placed a linear polarizer in front of the cmos camera to measure polarization distributions along 0°, 45°, 90°, and 135° directions, respectively. the emission pattern is nearly uniform across all directions (Fig. 5e), aligning well with the theoretical prediction (Fig. 2c).

Finally, we evaluate the phase vortex nature of the lasing emission using the self-interference technique. Specifically, we split the lasing beam evenly into two parts and overlap them in real space using a Mach-Zehnder interferometer setup. The resulting interference fringe

reveals a pair of oppositely directed forked pattern, as a hallmark of phase singularity in the wavefront. The dislocation points of these forks align with the center of laser beams, where a fringe splits into two distinct branches, confirming that the vortex beam carries an orbital angular momentum (OAM) of $|l| = 1$. Due to the intrinsic chirality of the twisted bilayer metasurface, the fork patterns consistently orient in the same direction, regardless of variations in the pump beam's position or shape. This indicates that the phase vortex's handedness remains highly stable, irrespective of external excitation. By fitting experimentally observed fringe contrast with the theoretical model[53], we estimate the observed purity of chirality to be approximately $a_{CW}: a_{CCW} = 0.7: 0.3$, suggesting that the lasing action happens near the EP condition as shown in Supplementary Fig. S5. Due to nanoscale deviations in the radius caused by etching process variations from the designed structure at the EP point, the performance deviates from the EP condition, consequently degrading the chirality purity. To verify that the observed vortex behavior intrinsically arises from the twisted-bilayer geometry, we fabricated single-layer reference samples that retain an achiral structure for comparison. In contrast to the single-layer samples, the twisted-bilayer structures consistently exhibit the same fork pattern and orientation across multiple samples and positions (see Supplementary Section 10 for further details). We also confirm that the vortex feature corresponds to a steady-state behavior rather than a transient response, based on carrier-dynamics analysis (Supplementary Section 11).

## Discussion

Photon's chirality is typically linked to its spin or saying polarization[56], manifesting as circularly polarized states. Our findings reveal that a twisted bilayer system can bend photon motion, exceptionally forming vortices in real space with orbital chirality. This effect is driven by rich helical and non-Hermitian coupling in Moiré lattice to enable collective oscillations. In the context of orbital chirality, a twisted bilayer system fundamentally differs from a single-layer Moiré lattice due to its inherent chirality and non-Hermiticity[24,48,57], which spontaneously lifts the reciprocity to distinguish left and right-handedness. As discussed in Supplementary Section 12, the built-in structural chirality of twisted-bilayer system itself enables active and intrinsically defined OAM generation directly in real space.

In an aspect of physics, our observation provides striking evidence of sophisticated lattices generating exotic collective behaviors in real space, paving the way for unexplored phenomena such as extraordinary vortex transportation and strong spin-orbit interactions of light. The collective oscillation in twisted photonics might also inspire new possibilities for collective electron behavior in bilayer graphene and other two-dimensional materials, hinting the importance of chirality and non-Hermiticity in long-range and strong correlation. From a technical perspective, the twisted bilayer metasurface presents a promising platform for realizing compact chiral light sources. Beyond the advantages on realizing stable and intrinsic orbital chiral lasing without external helical perturbation, the twisted bilayer structure can slow the photon's group velocity and cooperate with BICs, thus effectively suppressing lateral and vertical leakages for low threshold lasing. Additionally, the surface-emitting nature of twisted bilayer metasurfaces can enable considerable output power, which is essential for optical manipulation[58], detection[59], and communication[60].

We have reported the observation of intrinsic orbital chiral lasing in twisted bilayer metasurfaces. We have found that the isotropic geometry and dispersion of every individual metasurface can drive photons to collectively oscillate and rotate in either clockwise or counter-clockwise directions. Combined with helical and non-Hermitian couplings in the bilayer system, it breaks chiral symmetry spontaneously leading to orbital chiral emission at the non-Hermitian degeneracy upon complex bands. By developing a wafer bonding process to fabricate the sample, we have experimentally achieved stable single-mode lasing spanning a broad spectral range of 250 nm at the optical pump threshold of 73 kW/cm². The intrinsic orbital chiral nature of twisted bilayer metasurface has been verified by characterizing the localized doughnut lasing beam in real space that carries a phase vortex with nontrivial orbital angular momentum. Our findings suggest multiple opportunities in creating orbitally chiral photons in sophisticated lattices, thus enriching our understanding of twisted photonics and potentially paving the way for compact chiral light sources for versatile applications.

## Method
### Numerical simulations
The numerical simulations were performed using the finite-element method (FEM, COMSOL Multiphysics). The bandgap structure shown in Fig. 1b and the TE-A mode's field distribution depicted in Fig. 1c were calculated using a 3D unit-cell structure of a single-layer metasurface. This model employed Floquet periodic boundary conditions on the sidewalls and perfectly matched layers (PMLs) at the top and bottom boundaries. The results presented in Figs. 2 and 3c were obtained from calculations based on our designed Moiré superlattice structure. This computational model similarly utilized Floquet periodic boundary conditions on the sidewalls and PMLs at the top and bottom boundaries. The computational model was executed on a system equipped with two Intel(R) Xeon(R) E7-4890 v2 @ 2.80 GHz CPUs. Within the finite element model, the mesh discretization for the photonic crystal layer utilized the minimum element size of 3 nm and the maximum element size of 80 nm. Other key parameters for mesh generation include the maximum element growth rate of 1.45, the curvature factor of 0.5, and the resolution of narrow regions set to 0.6.

### Sample fabrication
The sample fabrication was performed using the low-temperature annealing direct-bonding processes via a designed high-precision alignment. Following lithography and Ti/Au sputtering processes, we lift off the metal windows of 200 μm × 200 μm on top of metasurfaces by acetone developing. The total thickness of Ti (15 nm)/Au (35 nm) is 50 nm and the thickness of the metals can be controlled precisely in sputtering processes. The sputtering metals on both sheets provide a total of 100 nm air gap, as an additional degree of freedom for inter-layer interactions. Both the auxiliary alignment marks and the periodic holes are patterned on each slab by EBL(ELX-F125), and subsequently the metasurfaces are etched by ICP. Selective dry etching of silicon dioxide is performed for 250 s under an inductive coupling power of 900 W, a RF bias power of 50 W, chamber pressure of 3.75 mTorr, and gas flow rates of 10/30 sccm for $CHF_3/CF_4$. Further, the metasurface on the InP epiwafers are fabricated using PlasmaPro 100 Cobra system (Oxford Instruments). A fast etch rate is achieved in 60 s under an inductive coupling power of 500 W, a RF bias power of 170 W, chamber pressure of 5 mTorr, gas flow rates of 12/2/30 sccm for $BCl_3/Cl_2/Ar$, and a temperature of 220 °C.

Following the etching process, the undercut InP epitaxial layers below their active regions are over-etched by 1: 1= 60%HCl: $H_2O$ solution in 11 min. Then the active die is picked-and-placed on the interposer die at the designed position by the dual-view microscope in flip-chip instrument via alignment marks. The alignment and prebonding processes are done under standard atmospheric pressure, 3 K/s heating rate, annealing temperature of 300 °C in 360 s, 2 K/s cooling rate, and 25 N pressure. Then the aligned and pre-bonded bilayers are moved into the Suss bonder machine immediately and gently. The fitted twisted bilayers are bonded based on a four-step bonding process under 0.0001 mbar atmospheric pressure. Initially, the temperature of the bonder elevates to 120 °C at a rate of 30 °C/min under no pressure, and the temperature is maintained at 120 °C in 20 min. Secondly, the bonder starts to bond under a pressure of 500 mbar in

10 min. Thirdly, the annealing temperature is enhanced to 150 °C at a rate of 10 °C/min and the annealing time is maintained at 60 min. At last, the bonder is cooled through natural cooling. The four-step bonding process is a low-pressure and low-temperature annealing method for bonding twisted bilayers, meanwhile, the thickness of TiAu can be maintained. Notably, this bonding process retains the fragile metasurface as bilayers, suspending in midair intact after multi-step bonding.

## Measurement and data processing

The experimental setup is based on a confocal microscopy system with a free-space laser as the pump light source. A height-adjustable mount is placed before the laser (MPL-N-1064-200uJ, 1064 nm, 10 kHz, 2 ns pulse) for precise alignment with the optical axis. A 20× Mitutoyo objective focuses the beam to 10 μm. The optical path includes lens L1 ($f$ = 200 mm), confocal with the objective, and a $4f$ system formed by L2 ($f$ = 250 mm) and L3 ($f$ = 300 mm), with L2 enabling near-to-far field switching. A 1300 nm long-pass filter blocks the pump light, allowing observation of lasing emission at 1550 nm. The measurement system includes an InGaAs infrared CMOS camera (IMX990, Sony) and a monochromator (IsoPlane SCT320 with NIRvana 640, Princeton Instruments) equipped with 600 g/mm and 150 g/mm gratings for spectral analysis.

The CMOS image data is processed by subtracting the background image to extract the lasing pattern. The background image, captured when the pump light is off, reflects the CMOS camera's idle response, including dark current, thermal noise, and stray light, and is used as a constant reference. During measurements, each image is corrected by pixel-wise subtraction of this background, effectively suppressing the camera-related noise. This enhances the signal-to-noise ratio, allowing weak features such as interference fringes and fork patterns to be clearly observed.

### Reporting summary

Further information on research design is available in the Nature Portfolio Reporting Summary linked to this article.

## Data availability

All data are available in the Article, its Supplementary Information, or Source Data file. Source data in Fig. 1 and Fig. 3 in this study are provided in Supplementary Dataset. All data are available from the corresponding author upon request. Source data are provided with this paper.

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

## Acknowledgements

This work was supported by National Key Research and Development Program of China (2022YFA1404804), National Natural Science Foundation of China (62205328, 62325501, 62135001 and 62405008, 12521004), Huawei Technologies Co. Ltd. Grant (TC 20220323035), Beijing Nova Program (20230484332), Australian Research Council (Grant No. DP210101292), and the International Technology Center Indo-Pacific (ITC IPAC) via Army Research Office (contract FA520923C0023). The simulation of this work was supported by the High-performance Computing Platform of Peking University.

## Author contributions

M.W., N.L. and Z.Z. contributed equally to this work. M.W., N.L., Z.Z. and C.P. conceived the idea. M.W., N.L., Z.Z., Y.C., X.Y., Z.L., C.P., W.Z. and Y.K. performed the theoretical study and simulation. M.W., N.L., Z.Z., Y.C., J.S., J.C., C.T.,D.X., Z.Y. and C.P. conducted the fabrication and measurement. M.W., N.L., Z.Z., Y.C., W.Z., C.P. and Y.K. wrote the manuscript with input from all authors. W.Z., Y.K. and C.P. supervised the research. All authors discussed the results.

## Competing interests

The authors declare no competing interests.
