## [Peer Review File · Nature Communications]

Orbital chiral lasing in twisted bilayer metasurfaces

Corresponding Author: Professor Chao Peng

Version 0:

Reviewer comments:

Reviewer #1

(Remarks to the Author)

This manuscript presented a twisted bilayer photonic system where two separate semiconductor membrane metasurfaces are bonded vertically with a twist. By optically pumping the above system, single-mode lasing with orbital chiral characteristics were observed. The experiments are of high technical quality. Nevertheless, more theoretical analyses, numerical calculations, and experimental results are expected to enrich this manuscript and help attract wider attention in the nanophotonics community. Several comments on this manuscript are:

1. While the effective Hamiltonian is useful, the absence of a complete bandstructure calculation for the bilayer system is disappointing. A supercell analysis would directly reveal the Bloch modes and their couplings, and its omission leaves the theoretical basis less convincing.
2. On page 4, the authors claimed that the design relies on a 620 nm membrane thickness with a 100 nm interlayer gap. However, throughout the manuscript, the authors did not provide an in-depth explanation on how these numbers were selected or optimized.
3. The coupling constants k^{intra} and k^{inter} appear chosen to reproduce the observed spectra rather than obtained from the structure itself. Without numerical extraction or experimental calibration, the model risks being descriptive rather than predictive.
4. The condition $k^{\text{inter}} = -k^{\text{intra}}$ is presented as defining an exceptional point. It would be useful to clarify whether this condition has physical meaning in the realized structure, or if it is primarily a mathematical construct introduced for interpretation.
5. The gain-guided cavity provides lateral confinement, yet its interaction with the twist-induced coupling is not explicitly considered. Treating these as separate effects simplifies the picture and leaves open the question of how the gain profile modifies the claimed chiral response.
6. The sample fabrication details of EBL and ICP should also be included in this manuscript.
7. OAM lasing has been experimentally demonstrated based on a variety of physics with diverse symmetry-breaking approaches and photonic cavity designs. Furthermore, one single photonic crystal slab is enough to generate OAM beams (for instance, Wang, B. et al. Nat. Photon. 14, 623–628 (2020).) Please specify the merits and advantages of the proposed twisted bilayer metasurfaces in the field of OAM generation.

Reviewer #2

(Remarks to the Author)

I co-reviewed this manuscript with one of the reviewers who provided the listed reports. This is part of a Nature Communications initiative to promote peer review training and provide appropriate recognition to early-career researchers who co-review manuscripts.

Reviewer #3

(Remarks to the Author)

The authors describe the observation of a chiral orbital laser from a twisted photonic structure fabricated using a wafer bonding process. The results demonstrate a low-threshold laser with a broad spectral range of 250 nm and also show a doughnut pattern in real space.

Experimental evidence and physical principles are strongly supported by numerical simulations and theoretical models that clarify some key concepts.

The sample realization is also well detailed as well as the experimental characterization. Finally the work is well supported by the literature.

For these reasons, I think the work is suitable for publication in Nature Communications.

just a curiosity: could there be contributions due to SAM?

Reviewer #4

(Remarks to the Author)

The paper is undoubtedly groundbreaking; however the authors should address some points to strengthen their work and its impact.

These points focus on the depth of evidence, mechanistic clarity, and the claim of intrinsic chirality. In particular, I found that there is a lack of direct comparison with experiments. The most significant weakness is the absence of a direct experimental comparison with a non-twisted bilayer or a single-layer metasurface. The central claim is that the twist itself is responsible for breaking the degeneracy and enabling intrinsic orbital chiral lasing. Without a control experiment, it is difficult to rule out that the observed chiral lasing isn't simply a result of the gain-guided cavity and the dispersion of the single layer, which could also support chiral modes, albeit in a degenerate and non-preferential manner. Demonstrating that a 0° twist bilayer produces a non-chiral or degenerate lasing output would make the role of the twist irrefutable.

The paper claims the chirality is intrinsic and stable, but the experimental data does not conclusively prove that the system selects a specific handedness due to the twist. Did all measured devices and pump spots produce a vortex with the same topological charge? Or was it random? If it's random, the intrinsic nature is questionable; the twist creates a chiral potential, but random defects or pump fluctuations might still determine the final state. If it's always the same, what is the physical mechanism in the structure that fixes the handedness? The estimated chirality purity of 70:30 is good but not exceptional. A major question is whether this ratio is fundamentally limited by the design or if it can be improved to near 100:0 by operating precisely at the EP.

The gain-guided mechanism, while clever, introduces a significant and complex variable. The achiral pump creates a chiral laser, which is a fantastic result, but it also makes it difficult to disentangle the pure twist-induced effects from nonlinear interactions between the gain, carrier diffusion, and the photonic structure. It remains unclear to me how much of the mode selection is due to the twist-induced non-Hermitian coupling versus gain competition in a complex, non-uniform medium. A numerical model that incorporates spatiotemporal gain dynamics would significantly strengthen the claim that the observed phenomenon is primarily a photonic (twist) effect rather than a gain-dynamics effect.

In conclusion, I believe the paper should be published after the authors have considered the above criticism which does not impact on the strength and quality of the results presented here.

My final recommendation is: publish after minor revisions.

Version 1:

Reviewer comments:

Reviewer #1

(Remarks to the Author)

The authors have addressed the comments. A few minor issues could be considered for further improvement of the final manuscript:

1. The discussion on the approximately 70:30 chirality purity ratio is insightful. The authors are encouraged to include this discussion in the main text and analyze some specific geometric parameters and/or fabrication issues that limit this chirality purity.

2. In the Sample Fabrication section, the material "GaInP" is mentioned. Please doublecheck this material name.

Reviewer #4

(Remarks to the Author)

I have carefully examined the revised manuscript and the authors' responses to the previous referee reports. The revision and the responses are substantial and convincing, and earlier concerns have been satisfactorily addressed.

In particular, the authors have significantly strengthened the theoretical framework, clarified the physical meaning of the effective Hamiltonian and coupling constants, and improved the discussion of design choices and optimization parameters. The authors have clearly placed the proposed twisted bilayer metasurface laser more clearly within the broader context of orbital-angular-momentum (OAM) photonics, and the authors articulate convincingly the advantages of the twisted bilayer approach compared to single-layer implementations. The discussion of chirality, exceptional points, and gain-guided confinement is balanced and physically transparent.

While some aspects of gain dynamics and chirality purity could be further explored in future works, these points no longer detract from the validity or impact of the present study.

In conclusion, I believe the manuscript is suitable for publication in Nature Communications, and I support its acceptance.

Reply to Reviewers' comments and a summary of the changes made in the revised manuscript

Response to Reviewer #1:

General Comments: This manuscript presented a twisted bilayer photonic system where two separate semiconductor membrane metasurfaces are bonded vertically with a twist. By optically pumping the above system, single-mode lasing with orbital chiral characteristics were observed. The experiments are of high technical quality.

Response: We thank the reviewer for his/her recognition and positive comments of our work.

General Comments: Nevertheless, more theoretical analyses, numerical calculations, and experimental results are expected to enrich this manuscript and help attract wider attention in the nanophotonics community. Several comments on this manuscript are:

Response: We appreciate the referees' careful review and have revised the manuscript following their suggestions. We hereby present the point-by-point replies to the reviewers' comments.

Comments 1: While the effective Hamiltonian is useful, the absence of a complete bandstructure calculation for the bilayer system is disappointing. A supercell analysis would directly reveal the Bloch modes and their couplings, and its omission leaves the theoretical basis less convincing.

Response: We sincerely thank the reviewer for this insightful comment. We agree that although the effective Hamiltonian offers a clear and intuitive description of the bilayer physics near the Γ point, a full band-structure calculation based on the supercell is essential for validating the origin of the Bloch modes and their couplings.

In response to this suggestion, we have performed full-wave eigenfrequency simulations (COMSOL) for the twisted bilayer supercell and included the results in the revised Supplementary Information (see Fig. R1 and related text).

The calculated supercell band structure (Fig. R1) reveals a complex spectral landscape induced by the Moiré pattern. As expected, the enlarged supercell reduces the Brillouin zone (BZ) into a mini-BZ, producing a dense set of bands through extensive zone-folding. Despite this complexity, the modes of primary interest, namely, the hybridized odd and even states originating from the fundamental TE-A band, remain clearly identifiable.

Specifically, the calculated supercell data confirms several facts:

- ◆ **Light line:** the outer line of light cone can be clearly observed, marked by dashed-lines in Fig. R1.

- ◆ **Zone folding:** the original photonic bands of the unit cell fold into the supercell's mini-BZ, giving rise to multiple bands that can be classified according to their specific folding pathways.
- ◆ **Mode hybridization:** interlayer coupling lifts the degeneracy of several folded bands, including the TE-A odd and even modes that are central to our lasing competition analysis.
- ◆ **Region of interest:** Although the overall spectrum is dense, the quadratic-dispersion region relevant to lasing remains clearly identifiable (highlighted by the red box). This confirms the validity of our effective Hamiltonian model, which focuses primarily on the high-Q branches of interest.

We believe this additional data provides the complete view of the supercell band structure as requested by the reviewer, and further confirms that the bands remain relatively isolated in the region of interest. This isolation justifies the use of our effective Hamiltonian to accurately capture the system's essential physics. We will elaborate further on the region of interest in our responses to the reviewer's additional comments.

Fig. R1 | Full band structure of the twisted bilayer Moiré supercell. Calculated eigenfrequency dispersion of the supercell along the high-symmetry directions of the mini-Brillouin zone. Owing to the large real-space periodicity of the Moiré pattern at a twist angle of 22.62° , the band structure exhibits a dense set of bands arising from zone-folding. Despite this complexity of folded bands, the targeted hybridized modes remain clearly identifiable within the frequency range of interest (red-box), thereby confirming the validity of the effective model in describing the specific lasing action.

Fig. R2 | Zoomed-in view of the region of interest shown in Fig.R1. A zoomed-in view of the band structure of the twisted-bilayer Moiré supercell within the red box highlighted in Fig. R1. Within the shaded region, the TE-A odd and even bands are clearly identified as the candidate lasing modes, originating from the hybridization of the TE-A bands of the upper- and lower-unit cells at the Γ point.

Comment 2: On page 4, the authors claimed that the design relies on a 620 nm membrane thickness with a 100 nm interlayer gap. However, throughout the manuscript, the authors did not provide an in-depth explanation on how these numbers were selected or optimized.

Response: We sincerely thank the reviewer for drawing our attention to this missing information.

The geometric parameters, specifically the membrane thickness and the interlayer air gap, were carefully designed to enable the experimental demonstration of orbital chiral lasing. The underlying design principles are summarized below:

- ◆ **The membrane thickness is optimized to promote the Q for lasing**

As shown in Figs. R1 and R2, our modes operate at the Γ point and are radiative. The choice of membrane thickness (620 nm, corresponding to 1.21 a) is crucial for achieving sufficiently high-Q factors to support efficient lasing. As detailed in Section 3 of the Supplementary Information, the Moiré lattice inevitably generates multiple diffractive orders that fall within the light cone, even with careful tuning of the supercell size.

At a membrane thickness of $h=1.21a$, the off- Γ BICs in momentum space become aligned in a way that suppresses radiation from multiple diffractive orders. As shown in Fig. S2 of the Supplementary Information, this optimization yields a theoretical Q exceeding 2.5×10^7 , which

is enough to support low-threshold lasing. Therefore, we chose the membrane thickness as $h=620\text{nm}$ ($1.21a$).

◆ **The interlayer air gap is optimized by balancing mode splitting and Q.**

The interlayer gap (100 nm) was chosen as a trade-off between achieving adequate mode frequency splitting and maintaining sufficiently high-Q factors for lasing. To clarify this point, we performed additional simulations of the supercell mode eigenfrequencies as a function of the air-gap thickness. In Fig. R3, we observe that the TE-A odd and even modes move closer in frequency as the air gap increases, indicating a weakening of the interlayer coupling strength. Therefore, the air gap cannot be too large, otherwise the two modes become insufficiently distinguishable.

Fig. R3 | The mode-splitting vs. air-gap. The TE-A odd and even modes (red and blue lines) approach near degeneracy when the air gap becomes too large.

Fig. R4 | The Qs vs. air-gap. The Qs of TE-A odd and even modes (red and blue lines) increases with air-gap.

On the other hand, the air gap cannot be too small, as this reduces the Q factors of the TE-A odd and even modes and decreases their contrast due to strong interlayer coupling, which is unfavorable for lasing. Additionally, an extremely small air gap poses practical fabrication challenges, since reliable metal deposition requires a minimum thickness of approximately 50 nm to ensure flatness and high quality. As a balance, we choose air-gap as 100 nm.

Comments: 3. The coupling constants κ^{intra} and κ^{inter} appear chosen to reproduce the observed spectra rather than obtained from the structure itself. Without numerical extraction or experimental calibration, the model risks being descriptive rather than predictive.

Response: We thank the reviewer for this insightful comment.

We acknowledge that the coupling constants κ^{intra} and κ^{inter} presented in the original manuscript are somewhat phenomenological and cannot be directly calculated from the structural parameters.

However, we here clarify that the effective Hamiltonian is not merely a fitting model; it can be derived from electromagnetic perturbation theory. As a result, the coupling coefficients can be calculated from the overlap integrals of the mode profiles and the dielectric perturbation introduced by the twisted bilayer geometry.

Briefly, we employ electromagnetic perturbation theory on Maxwell's equation, the generic form of coupling coefficients κ_{ij} between mode i (with field \mathbf{E}_i and mode j (with field \mathbf{E}_j) is given by the overlap integral with dielectric perturbation $\Delta\epsilon$:

$$\kappa_{ij} \approx \frac{\omega_0}{2} \frac{\int_V \Delta\epsilon(\mathbf{r}) \mathbf{E}_i^*(\mathbf{r}) \cdot \mathbf{E}_j(\mathbf{r}) dV}{\int_V \epsilon(\mathbf{r}) |\mathbf{E}_i(\mathbf{r})|^2 dV}$$

♦ **Intra-layer Cross-Coupling κ^{intra}**

For the intra-layer coupling (e.g., coupling between CW and CCW modes within the lower layer triggered by the upper layer), $\Delta\epsilon$ corresponds to the dielectric grating of the upper metasurface. Since the upper metasurface is rotated by a twist angle θ , the perturbation function follows the lattice periodicity. The coupling strength is determined by:

$$\kappa_{1,2}^{intra} \propto \int_{V_{upper}} \Delta\epsilon_{upper}(\mathbf{r}, \theta) \mathbf{E}_{CCW}^{l,*}(\mathbf{r}) \cdot \mathbf{E}_{CW}^l(\mathbf{r}) dV$$

♦ **Inter-layer Coupling κ^{inter}**

Similarly, the inter-layer coupling involves the overlap of the evanescent tails of the modes in the gap region and within the material of the opposite layer. For instance, the direct coupling is:

$$\kappa_{direct}^{inter} \propto \int_V \Delta\epsilon_{gap}(\mathbf{r}) |\mathbf{E}_{CW}(\mathbf{r})|^2 dV$$

The non-Hermitian coupling coefficients can also be derived, in which $\Delta\epsilon$ is introduced by the adjacent layer, as:

$$\kappa_1^{inter} = \frac{\omega_0}{2} \int \mathbf{E}_{CCW}^{u,*}(\mathbf{r}) \cdot \Delta\epsilon(\mathbf{r}) \cdot \mathbf{E}_{CW}^l(\mathbf{r}) dV$$

As a result, the coupling coefficients can be in principle calculated from the modes' profile $E(r)$ and dielectric perturbation $\Delta\epsilon$.

However, given the complexity of the twisted bilayer system, a full-wave simulation on the entire supercell system trapped in the effective gain cavity is extremely computationally heavy, it is not quite feasible to directly retrieve accurate values of coupling coefficients. Instead, we try our best to extract the values from numerical simulation and experimental observations.

First, we note the splitting of even and odd modes is dominated by the direct inter-layer coupling strength κ_{direct}^{inter} . From the splitting value in Fig. R2 and R3, we observe a normalized frequency difference of $\Delta(a/\lambda) \approx 9 \times 10^{-5}$ between the split modes near the normalized frequency. This splitting magnitude corresponds to a direct inter-layer coupling strength κ_{direct}^{inter} on the order of 2.6×10^{-4} (in normalized units).

Second, we observe slight splitting of modes in the before lasing condition as 0.23 nm. According to our theory, this tiny split reflects the strengths of interplay strength of inter and intra-layer chiral coupling. This experimentally observed wavelength splitting corresponds to a chiral coupling strength κ^{intra} on the order of 10^{-4} .

Fig. R5 | Spectrum before lasing.

Comment 4: The condition $\kappa^{inter} = -\kappa^{intra}$ is presented as defining an exceptional point. It would be useful to clarify whether this condition has physical meaning in the realized structure, or if it is primarily a mathematical construct introduced for interpretation.

Response: We thank the reviewer for the question.

We clarify that the condition $\kappa^{inter} = -\kappa^{intra}$ is not merely a mathematical construct but holds a concrete physical significance rooted in the interference of scattering channels in the twisted bilayer system. In physics, the "intra-layer" and "inter-layer" cross-couplings originate from the same geometry, namely the dielectric perturbation introduced by the adjacent, twisted metasurface. However, they represent two distinct scattering processes:

Intra-layer coupling (κ^{intra}) describes the scattering of a mode into its counter-propagating counterpart within the same layer, mediated by **reflections** from the adjacent layer.

Inter-layer coupling (κ^{inter}) describes the scattering of the mode into the counter-propagating partner in the other layer, mediated by **evanescent tunneling**.

The condition $\kappa^{\text{inter}} = -\kappa^{\text{intra}}$ corresponds to a point of destructive interference between the two scattering pathways. At the EP, the rate of interlayer chirality transfer precisely balances the rate of intralayer chirality mixing in a non-Hermitian fashion. In our structure, this condition is physically accessible and can be controlled through geometric parameters, specifically the vertical interlayer gap and the twist angle.

Furthermore, regarding the number of modes, while the effective Hamiltonian is formulated on a 4×4 basis (CW/CCW combined with upper/lower layers), the strong direct inter-layer coupling splits the system into two energetically separated branches (even/odd supermodes). The EP we discussed occurs on the high-Q branch for lasing. Thus, in physics, the EP represents the coalescence of the two chiral supermodes on the lasing band, forcing the entire bilayer system to lock into a single, unidirectional rotating state ($V = [0,1,0,1]^T$).

We have revised the manuscript and Supplementary accordingly to include our derivation and interpretation on couplings coefficients, thereby bridging the mathematical model to the physics of the twisted bilayer metasurface.

Comments 5: The gain-guided cavity provides lateral confinement, yet its interaction with the twist-induced coupling is not explicitly considered. Treating these as separate effects simplifies the picture and leaves open the question of how the gain profile modifies the claimed chiral response.

Response: We thanks for this insightful comment regarding the interplay between the gain-guided confinement and the twist-induced coupling.

In our theory, we employ a hierarchical perturbation approach, valid when the macroscopic gain profile varies slowly relative to the lattice constant.

First, we consider the gain-guided cavity effect. The spatially localized optical pump, with a Gaussian profile, introduces a radially dependent imaginary permittivity contrast. This forms a soft-boundary confinement that discretizes the membrane's continuum modes into discrete resonances. Owing to the isotropic nature of the pump spot, rotational symmetry is preserved, and the resulting eigenstates are well described by Laguerre-Gaussian wavefunctions with quantized radial (p) and azimuthal (l) indices as our basis.

Second, we introduce the structural twist, treating the Moiré lattice potential and interlayer interactions as a perturbation acting on the previously defined basis. The interaction between the gain profile and the twist coupling is implicitly included in the coupling coefficients, as the mode profile is shaped by the gain. Since the pump profile is geometrically isotropic, it does not introduce additional chiral symmetry breaking terms.

Similar method had been employed in literature [Prof. Fan's group, Phys. Rev. Lett. 126, 136101 2021], modeling the rest layer as a homogeneous, achiral dielectric slab to study wave interaction in one layer. In contrast, our method models the gain profile as an isotropic, slow-varying profile.

Comment 6: The sample fabrication details of EBL and ICP should also be included in this manuscript.

Response: Thanks for the question.

The sample fabrication relies on electron-beam lithography (EBL) for precise alignment of metasurface membranes and high-aspect-ratio inductively coupled plasma (ICP) etching for the InP-based epi-wafer. A silicon dioxide hard mask is first deposited on the substrate via plasma-enhanced chemical vapor deposition (PECVD), prior to spin-coating a 500 nm thick layer of Zep-520A photoresist at 4000 rpm. Following a softbake, the epiwafers are exposed by EBL(ELX-F125) under the following parameters: beam energy of 125 keV, beam spot size of 2 nm, exposure dose of 380 $\mu\text{C}/\text{cm}^2$.

After developing with fresh developer and undergoing a series of clean processes, the silicon dioxide hard mask are fabricated via ICP dry etching. Selective dry etching of silicon dioxide is performed for 250 seconds under an inductive coupling power of 900 W, a RF bias power of 50 W, chamber pressure of 3.75 mTorr, and gas flow rates of 10/30 sccm for CHF₃/CF₄.

Further, the metasurface on the InP epiwafers are fabricated using PlasmaPro 100 Cobra system (Oxford Instruments). A fast etch rate is achieved in 60 seconds under an inductive coupling power of 500 W, a RF bias power of 170 W, chamber pressure of 5 mTorr, gas flow rates of 12/2/30 sccm for BCl₃/Cl₂/Ar, and a temprature of 220 °C.

Comment 7: OAM lasering has been experimentally demonstrated based on a variety of physics with diverse symmetry-breaking approaches and photonic cavity designs. Furthermore, one single photonic crystal slab is enough to generate OAM beams (for instance, Wang, B. et al. Nat. Photon. 14, 623–628 (2020).) Please specify the merits and advantages of the proposed twisted bilayer metasurfaces in the field of OAM generation.

Response: We appreciate the reviewer for raising this the important point regarding the unique merits of our proposal.

The reviewer is correct that a wide range of photonic structures has been employed for OAM generation. To keep our response concise and focused, we categorize the existing design principles into three classes—**local**, **non-local**, and **collective**. Within each class, we further review the approaches from multiple perspectives, including whether they are **passive or active**, **real-space or momentum-space**, and **extrinsic or intrinsic**.

♦ **Local design.**

This principle is widely used in metasurfaces or waveplate, where wave interactions occur in a tight-binding manner. In such structures, OAM is generated directly and locally through geometric variations. These devices are typically excited by external light sources and often combine Pancharatnam-Berry (PB) phases, offering substantial flexibility for tailoring wave fronts, including OAM generation. However, this class of structures is less suitable for lasing due to their relatively low Q factors. As a result, they are typically used for **passive**, **real-space**, and **extrinsic** OAM generation rather than active lasing applications.

♦ **Non-Local design.**

This principle is typically implemented in bulk photonic crystal (PhC) slabs, where the modes are long-range correlated and wave front manipulation takes place in **momentum space**, often in conjunction with PB phases near polarization singularities. Such systems can operate in either

passive (reflective/transmissive) or **active** (lasing) manner because they support sufficiently high Q factors. However, they generally require external waveplates to prepare the desired input polarization, and therefore they are classified as **extrinsic** OAM generators. The work referenced by the reviewer [Wang, B. et al., Nat. Photon. 14, 623–628 (2020)] falls into this category.

♦ **Collective design.**

This principle represents a hybrid of local and non-local design mechanisms. In the local picture, each site couples to a set of neighboring sites in real space, giving rise to collective lattice resonances. Alternatively, in the non-local picture, discrete guided resonances in momentum space can hybridize into collective modes. Consequently, the wave front can be engineered in both real and momentum space, enabling desired OAM generation. Circular PhC cavities with collective guided resonances (CGRs) [Nature Nano. 20, 1205–1212 (2025)], as well as PhCs with dislocations or disclinations, fall within this category. This class supports both **passive** and **active** operation and allows OAM generation in either **real-space** or **momentum-space**. The design in this work belongs to this category.

It is worth noting that the OAM generation in our work is uniquely **intrinsic**, arising directly from the inherent chirality of the twisted-bilayer system. Physically, stable OAM generation requires distinguishing between clockwise (CW) and counterclockwise (CCW) rotating states. While this separation can be achieved through external control of the pump profile, our design leverages the built-in structural chirality itself, enabling **active** and **intrinsically** defined OAM generation directly in **real space**.

It is difficult to comment on which method is more advantageous, as each may be suited to different application scenarios. Our design demonstrates that intrinsic structural chirality can facilitate OAM generation by leveraging exceptional points (EPs) and tailoring the modal Q-factors for lasing. This introduces a new addition to the OAM-generation toolbox, offering a compact and stable route toward intrinsically chiral OAM lasers.

Response to Reviewer #2:

General Comments: I co-reviewed this manuscript with one of the reviewers who provided the listed reports. This is part of a Nature Communications initiative to promote peer review training and provide appropriate recognition to early-career researchers who co-review manuscripts.

Response: We thank the reviewer for his/her recognition and positive comments of our work.

Response to Reviewer #3:

General Comments: The authors describe the observation of a chiral orbital laser from a twisted photonic structure fabricated using a wafer bonding process. The results demonstrate a low-threshold laser with a broad spectral range of 250 nm and also show a doughnut pattern in real space.

Experimental evidence and physical principles are strongly supported by numerical simulations and theoretical models that clarify some key concepts.

The sample realization is also well detailed as well as the experimental characterization. Finally the work is well supported by the literature. For these reasons, I think the work is suitable for publication in Nature Communications.

Response: We thank the reviewer for his/her recognition and positive comments of our work.

Comment 1: just a curiosity: could there be contributions due to SAM?

Response: Thanks for this insightful question.

Spin angular momentum (SAM) manifests as a polarization vortex, whereas orbital angular momentum (OAM) is characterized by a phase vortex. It is true that SAM can also induce phase variations when PB phases associated with different polarization states are projected onto the wavefront, a detail discussed in our previous work [Nat. Nanotechnol. 20, 1205–1212 (2025)], Supplementary Section 9. For the reviewer's convenience, we have added the relevant figure here.

To isolate the contribution of the SAM, a polarizer was incorporated into the optical path of our measurement setup to suppress the phase discontinuity from polarization vortex, thereby ensuring that the observed vortex features originate solely from OAM.

Figure S11: **Self-interference patterns of phase, polarization vortices, and their combination** (a) The principle of interference between two vectorial (polarization) vortices, in which an extra phase difference is induced due to their vectorial nature. (b) The field intensity of E_x , E_y , and total intensity for the polarization vortex (upper panel), phase vortex (mid panel), and their combination as ring vortex (lower panel), respectively.

Fig. S11 from Supplement Information [Nat. Nanotechnol. 20, 1205–1212 (2025)].

Response to Reviewer #4:

General Comments: The paper is undoubtedly groundbreaking; however the authors should address some points to strengthen their work and its impact.

Response: We thank the reviewer for his/her recognition and positive comments of our work.

Comment 1: These points focus on the depth of evidence, mechanistic clarity, and the claim of intrinsic chirality. In particular, I found that there is a lack of direct comparison with experiments. The most significant weakness is the absence of a direct experimental comparison with a non-twisted bilayer or a single-layer metasurface. The central claim is that the twist itself is responsible for breaking the degeneracy and enabling intrinsic orbital chiral lasing. Without a control experiment, it is difficult to rule out that the observed chiral lasing isn't simply a result of the gain-guided cavity and the dispersion of the single layer, which could also support chiral modes, albeit in a degenerate and non-preferential manner. Demonstrating that a 0° twist bilayer produces a non-chiral or degenerate lasing output would make the role of the twist irrefutable.

Response: We sincerely thank the reviewer for this valuable comment on strengthening our manuscript.

To verify that the twisted bilayer metasurface intrinsically possesses chirality for OAM, we followed the reviewer's suggestion and fabricated single-layer samples for comparison. We chose single-layer samples rather than untwisted bilayers not only because they are less fabrication heavy, but they best preserve the achiral nature of the structure and avoid unintended twisting during bilayer alignment. We keep the same structural parameters (air-hole period and radius) as the bilayer metasurface. The SEM pictures of samples are presented below:

Fig. R6 | SEMs for single-layer sample. The over view of metasurface with cleaved side-walls and the top view.

We fabricated three single-layer samples for comparison. Because the excitation light spot is smaller than the sample footprint, we performed measurements at two distinct excitation positions on each sample to ensure repeatability.

A representative lasing spectrum is shown in Fig. R7, showing almost one single peak across 20 nm wavelength range. A tiny peak appears adjacent to the main peak in the short-wavelength end, which may be due to the residual weak chiral symmetry breaking due to fabrication imperfection.

Fig. R7 | Lasing spectrum of single layer sample.

Furthermore, we performed self-interference measurements on the single-layer samples. To ensure that any potential fork features would not be overlooked, we expanded the field of view during imaging, the results are presented in Fig. R8. Across all three samples and two excitation positions per sample, no fork patterns were observed. This confirms that the lasing modes in the single-layer structures are non-chiral.

Fig. R8 | Self-interference of single layer samples. The interference patterns are presented for 3 samples each at 2 positions, no fork features were observed.

For comparison, we measured two additional bilayer samples at two different excitation positions, as shown in Fig. R9. In all cases, clear fork patterns were consistently observed. This confirms that the structural twist is indeed responsible for enabling orbital chiral lasing.

Fig. R9 | Self-interference of twisted samples. The interference patterns are presented for 2 extra samples each at 2 positions, two-branch fork features were consistently observed with the same orientation.

Comment 2: The paper claims the chirality is intrinsic and stable, but the experimental data does not conclusively prove that the system selects a specific handedness due to the twist. Did all measured devices and pump spots produce a vortex with the same topological charge? Or was it random? If it's random, the intrinsic nature is questionable; the twist creates a chiral potential, but random defects or pump fluctuations might still determine the final state. If it's always the same, what is the physical mechanism in the structure that fixes the handedness? The estimated chirality purity of 70:30 is good but not exceptional. A major question is whether this ratio is fundamentally limited by the design or if it can be improved to near 100:0 by operating precisely at the EP.

Response: We thank the reviewer's for the insightful question.

As shown in Fig. R9, we measured the fork patterns in two additional bilayer samples at two different excitation positions. Together with the results presented in Fig. 5 of the main text, we found that all observed fork patterns consistently exhibit two branches oriented in the same direction. This consistency indicates that the emitted vortices possess a well-defined and identical topological charge, confirming the deterministic and stable chiral lasing behavior enabled by our design.

The reviewer is correct that, although the twist introduces a chiral potential, random defects or pump-induced fluctuations could in principle influence the final lasing state if the twist-induced chirality were insufficiently strong. Our experimental results, however, indicate that the structural chirality provided by the twist is robust enough to consistently enforce the same topological charge. As detailed in Responses 3 and 4, we explicitly derive the coupling coefficients and clarify their physical meanings. Because lasing occurs on the most high-Q branch in the vicinity of the exceptional point (EP), the stability ('hardness') of the chiral state is primarily governed by the coupling scenario, which is deterministically set by the bilayer geometry rather than by stochastic perturbations.

Regarding the upper bound of chirality purity, theory indicates that pushing the lasing mode closer to the exceptional point (EP) monotonically improves the chiral selectivity. In principle, achieving arbitrarily high purity is not limited by theory but by technical constraints. Approaching the EP requires finely tuned coupling coefficients, which demands sophisticated engineering. While our work demonstrates that intrinsic chirality in a twisted bilayer system is effective, we acknowledge that the bilayer platform remains relatively complex from a practical design and fabrication standpoint. A promising future direction is to explore structurally simpler architectures that naturally possess intrinsic chirality, enabling more accessible implementations.

In addition, combining external and in-situ control of the pumping conditions may be necessary for practical vortex-laser implementations. For instance, electrically pumped lasers could address this challenge more effectively through optimized electrode layouts. In particular, asymmetric electrode designs may further enhance chiral purity by preferentially reinforcing one rotational channel over the other.

Comment 3: The gain-guided mechanism, while clever, introduces a significant and complex variable. The achiral pump creates a chiral laser, which is a fantastic result, but it also makes it difficult to disentangle the pure twist-induced effects from nonlinear interactions between the gain, carrier diffusion, and the photonic structure. It remains unclear to me how much of the mode selection is due to the twist-induced non-Hermitian coupling versus gain competition in a complex, non-uniform medium. A numerical model that incorporates spatiotemporal gain dynamics would significantly strengthen the claim that the observed phenomenon is primarily a photonic (twist) effect rather than a gain-dynamics effect.

Response: Thanks for the review's valuable comments.

The reviewer is correct that gain complexity may introduce additional effects beyond those induced by twisting, such as nonlinear carrier dynamics, diffusion, and thermal processes, such phenomena that are unavoidable in practical laser devices, especially in electrically pumped implementations. In this work, we treat the gain profile perturbatively, as detailed in our response to Comment 5 of Reviewer 1, in which we assume the gain medium is achiral and linear for simplicity.

The reviewer raises an insightful question regarding the temporal response of our system. Indeed, the spatiotemporal behavior of light has recently become a focal topic in photonic communications. It is important to verify whether our system exhibits any significant gain dynamics. To address this, we performed numerical simulations using the material parameters of the carriers.

We consider the rate equation for the carrier density N inside the active region (InGaAsP MQWs), which is given by

$$\frac{dN}{dt} = \frac{\eta J}{q d_{active}} - (R_{sp} + R_{nr}) - \sum_m v_{gm} g_m N_{pm}$$

where η is the optical efficiency of our laser, J is the current density, q is the elementary charge, R_{sp} is the spontaneous recombination rate, R_{nr} is the nonradiative recombination rate, v_g is the group velocity of the mode, g is the gain, N_p is photon density, and a separate photon density for each mode indexed by the interger m . For simulations, the temporal change curve of carrier density is calculated by solving equations:

$$\frac{dN_{pm}}{dt} = \left[\Gamma_m v_{gm} g_m - \frac{1}{\tau_{pm}} \right] N_{pm} + \Gamma_m R'_{spm}$$

For simplicity, in this study, we can approximate the gain spectrum by

$$g(N) = \frac{g_{max} * (N - N_{tr})}{N + g_{max}/abs(g_0) * N_{tr}}$$

TABLE I. Parameters used for simulations.

Symbol	Parameter	Value	Source
J	Current density	2e7 A/m ²	
d_{active}	Total thickness of Quantum wells	45 nm	
τ_{sp}	Carrier lifetime	1.5 ns	
Γ	Optical confinement factor	31.8%	Calculated by the TMM
n_g	Group refractive index	2.94	Calculated by the TMM
g_{max}	Maximum gain	1000 cm ⁻¹	
N_{tr}	Transparency carrier density	1.5e24 m ⁻³	
g_0	Absorption coefficient	-5000 cm ⁻¹	
α_{in}	Internal loss of materials	5 cm ⁻¹	
β	Spontaneous emission factor	1e-4	

As can be seen from this curve, the carriers are in a steady state at 2 ns. Therefore, for a pulse duration of 2 ns, the system can be considered to operate in a steady state, without exhibiting any nontrivial temporal response.

Fig. R10 | Carrier dynamics of laser sample. The system reaches a steady state of carrier population before 2 ns, thus operating without exhibiting nontrivial temporal response for pulse durations matching or exceeding this timescale.

Comment 4: In conclusion, I believe the paper should be published after the authors have considered the above criticism which does not impact on the strength and quality of the results presented here.

My final recommendation is: publish after minor revisions.

Response: We sincerely appreciate the reviewer's recognition of our work.

Response to Reviewers' Comments

Manuscript Title: Chiral orbital lasing in a twisted bilayer metasurface

Manuscript ID: NCOMMS-25-66090A

Corresponding Author: Chao Peng (pengchao@pku.edu.cn)

General Response: We deeply appreciate the reviewers' positive and supportive feedback, which is a great encouragement and motivation for us to continue pursuing rigorous and high-quality research.

Response to Reviewer #1:

Comments for the Author: The authors have addressed the comments. A few minor issues could be considered for further improvement of the final manuscript:

1. The discussion on the approximately 70:30 chirality purity ratio is insightful. The authors are encouraged to include this discussion in the main text and analyze some specific geometric parameters and/or fabrication issues that limit this chirality purity.
2. In the Sample Fabrication section, the material "GaInP" is mentioned. Please doublecheck this material name.

Response: We thank the reviewer for his/her recognition and positive comments on our work.

1. We add the analysis about fabrication issues that limit this chirality purity in the maintext. Due to nanoscale deviations in the radius caused by etching process variations from the designed structure at the EP point, the performance deviates from the EP condition, consequently degrading the chirality purity.
2. We sincerely thank the reviewer for pointing out this misleading phrase. We have corrected the material name to InP in the sample fabrication process.

Response to Reviewer #4:

Comments for the Author: I have carefully examined the revised manuscript and the authors' responses to the previous referee reports. The revision and the responses are substantial and convincing, and earlier concerns have been satisfactorily addressed.

In particular, the authors have significantly strengthened the theoretical framework, clarified the physical meaning of the effective Hamiltonian and coupling constants, and improved the discussion of design choices and optimization parameters.

The authors have clearly placed the proposed twisted bilayer metasurface laser more clearly within the broader context of orbital-angular-momentum (OAM) photonics, and the authors articulate convincingly the advantages of the twisted bilayer approach compared to single-layer

implementations. The discussion of chirality, exceptional points, and gain-guided confinement is balanced and physically transparent.

While some aspects of gain dynamics and chirality purity could be further explored in future works, these points no longer detract from the validity or impact of the present study.

In conclusion, I believe the manuscript is suitable for publication in Nature Communications, and I support its acceptance.

Response: We sincerely appreciate the reviewer's recognition and encouraging remarks regarding our work.